# Development of Mechanisms for Automatic Correction of Industrial Complex Tools in the Preprocessing of Laser Welding for Small-Scale and Piece Production Using Computer Vision †

**Rodionov Dmitry \*, Lyukhter Alexander and Prokoshev Valery**

Department of Physics and Applied Mathematics, Vladimir State University, 600000 Vladimir, Russia; 3699137@mail.ru (L.A.); prokoshev_vg@vlsu.ru (P.V.)

\* Correspondence: d3391rod@gmail.com; Tel.: +7-492-247-96-18

† This paper is an extended version of our paper published in "Dmitry Rodionov, Lyukhter Alexander, Prokoshev Valery, Methods of Automatic Correction of Technological Trajectory of Laser Welding Complex by Means of Computer Vision". In Proceedings of the 2020 IEEE International Conference on Industrial Engineering, Applications and Manufacturing (ICIEAM), 18–22 May 2020.

**Abstract:** This paper is devoted to the development and improvement of mechanisms for the functioning of an automated system for correcting the points of the planned trajectory of the tool of a laser robotic welding complex in the pre-process. Correction of the points of the planned trajectory is carried out in two stages: in the first stage, the focal laser radiation is corrected; in the second stage, the position of the tool is corrected. Correction of the focal laser radiation is carried out in conjunction with the automated focusing of the camera by moving the tool of the industrial complex along its own axis. The functioning of position correction mechanisms is based on methods for recognizing the edges of the gap line to be welded from the image obtained from the charge-coupled device (CCD) camera. The edges of the gap to be welded in the image are segmented using threshold selection. The boundaries of the threshold selection segment are the extreme values of the pixel distribution of the entire image in brightness. For unambiguous recognition of the desired edges based on the segmentation result, the features defining them as a pair of continuous, conditionally parallel lines are formalized. Relative to the recognized pair of edges, the correct position of the planned trajectory point relative to the current position of the welding head is determined. To transfer the correct position, we formalized the calculation model and chose an arbitrary point in the flat image in the workspace laser robotic welding complex, considering the orientation of the tool and the position of the camera. The results obtained made it possible to develop a correction system and successfully test it in the industrial complex.

**Keywords:** laser welding; remote welding; automated control system; industrial automation; technological preparation of production; production automation; computer vision; image processing

---

## 1. Introduction

The technological process of laser welding is one of the most promising high-tech methods for obtaining permanent joints of metal structures [1]. The use of compact fiber lasers for obtaining welded joints of metal products has considerable advantages [2] and significantly expands the design and technological capabilities of enterprises [3,4].

Robotic manipulators are used to deliver focused radiation to the treatment area. Combining technological and auxiliary welding equipment with a robotic arm forms a laser robotic welding complex (LRC-W).

However, the implementation of the LRC-W in small-scale and piece production is difficult. First of all, this is due to the high labor intensity of the production changeover and debugging of the technological solution [5]. As a result, the final cost of production increases significantly [6] and the introduction of laser welding into production becomes less expedient [7].

One of the ways to increase the efficiency of small-scale production is to increase the level of automation [8] of production at LRC-W [9]. Currently, enterprises with small-scale production are adapting the approaches of technological preparation used in large-scale production. However, the adaptation does not provide an effective reduction in the complexity of production changeover and debugging of technological solutions.

Technological preparation of robotic industrial complexes (process design, operation planning, selection of optimal equipment, creation of control programs, etc.) [10] is carried out in specialized computer-aided manufacturing (CAM) [11] and computer-aided process planning (CAPP) systems based on three-dimensional models [12]. The tool trajectory is transferred from the model space to the working space of the industrial LRC-W by using mathematical methods of coordinate transformation [13].

In contrast to ideal objects, which are three-dimensional geometric models built via computer-aided design (CAD), real parts may have some geometric deviations [14], so the laser beam trajectory built on a three-dimensional model can pass by the joint of parts, even in the case of an ideal interface of coordinate systems according to which the robot's control program was created.

Due to the fact that laser welding creates aggressive conditions for monitoring and measuring equipment, correcting the trajectory points on the fly is less preferable during execution.

The most promising solution to eliminate the problems of mismatch points of the planned trajectory is a software and hardware correction complex that functions in the preprocessing and automatically adapts the control program of movement to the installed blanks of the product in the tooling.

## 2. Problem Formulation

Deviation of the $p_{tool}$ position of focused laser radiation by a fraction of a millimeter during laser welding can significantly affect the quality of the result (Figure 1).

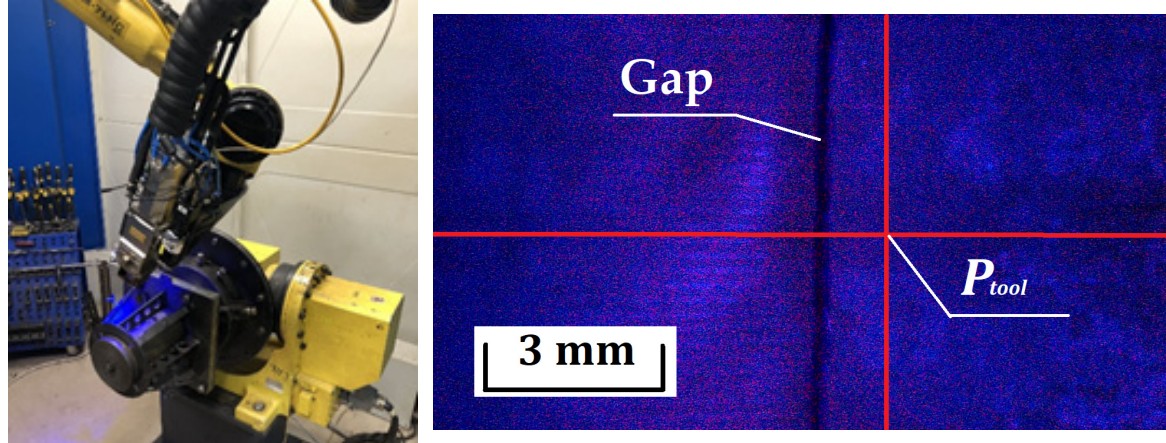

(**a**) LRC-W tool in working position.  (**b**) Photo of the position of the $p_{tool}$ tool obtained from the CCD camera installed on the LRC-W tool.

**Figure 1.** Positioning of the LRC-W tool in the process of planning a laser tack with an inaccurately located point of the trajectory relative to the gap between workpieces.

Today, software and hardware systems for correcting the points of the trajectory of the LRC-W tool are based on detecting the welded line [15]. In most cases, detection and recognition is performed using the following tools:

(1)  structured light tracking systems that project laser bands onto the study area, detecting and processing surface-distorted data [16] (for example, Meta Vision, Servo Robot, Scout, and Scansonic solutions);

(2)  software and hardware complexes for processing images obtained from a CCD camera of various configurations and operation spectra.

Structured light tracking systems allow good detection of the junction lines of workpieces with a significant gap but are poorly applicable in the case of small size [15]—that is, gaps that in most cases are welded by the LRC-W. The requirements of a fixed sensor location additionally impose restrictions on the positioning and orientation of the tool at the point of the projected trajectory when planning a preliminary detour. In addition, tracking devices are more expensive than solutions based on computer vision technologies.

Software and hardware complexes for processing images obtained with CCD do not have all the disadvantages of structural light tracking systems. However, for their stable operation, they require obtaining a high-quality image that allows them to sufficiently recognize the studied geometric primitives and outlines.

The first problem that occurs during recognition is a blurry image obtained as a result of the deviation of the LRC-W tool from the planned position. The result of the deviation is the defocusing of the CCD camera at the points of the translated trajectory from the CAM to the LRC-W workspace. Image blurring can be eliminated by automated focusing by moving the tool along its own orientation axis (in our case, along the axis of the laser beam). In other words, the problem is solved by developing an autofocus system for a CCD camera that uses the movement of an industrial complex tool as a focusing mechanism.

The second problem with obtaining a high-quality image is the presence of noise. As a solution to this problem, the research area is illuminated with LED illumination. For this purpose, a remote [17] bilateral LED illumination system was installed on the LRC-W tool (Figure 2).

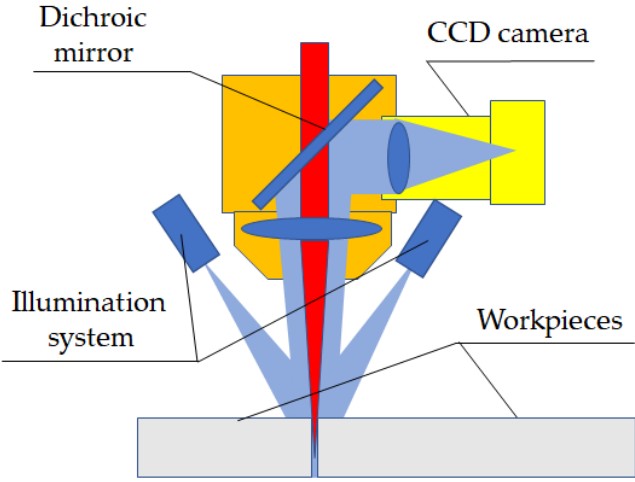

**Figure 2.** Scheme of the LRC-W tool.

To ensure that the points of the planned trajectory can be corrected relative to the geometric primitives and outlines of the processed part, the design data structure is expanded with data on the necessary orientation of the LRC-W tool at the correction point, which ensures a sufficient position of the CCD camera in the research area. In other words, in the process of designing a technological process in CAM, the process engineer additionally designs the position of the tool, which he will take

during the correction process, sufficient for recognizing geometric primitives and outlines, as well as correction of the position of the tool. Designing the position of the tool at the points of the trajectory will ensure the applicability of recognition methods for cases of non-trivial connections (angular, overlapping, t-shaped, and others).

Various solutions were identified to highlight the gap in the image obtained from the CCD camera, but most of them have disadvantages that do not allow them to be fully used as ready-made solutions for use in their own correction system. In this regard, there is a natural need to develop and improve existing methods and algorithms.

In [18,19], the gap is segmented by applying a median filter with a $3 \times 3$ transformation core size. However, the study does not specify the limitations and limits of applicability of the proposed method of segmentation of the gap. A similar conclusion can be drawn from [20], in which a threshold transformation is performed to select the gap, while the threshold value is not explicitly defined.

In [21,22], the gap is considered as a region of pixels with a lower gray level, allowing one to uniquely identify and segment the desired area. Unfortunately, they do not provide methods for determining the threshold selection for unambiguous line selection. Similarly, it is proposed to detect the gap in [23], where the selection of the desired line is performed by lightening the entire image until the main background becomes white, as a result of which only the gap is represented on the image. The question of determining the value of the lightening coefficient sufficient for contrast selection in the work remains open.

In [24], there is recognition of the gap between workpieces during welding. The visibility of the desired object is achieved by using two-way illumination, which allows one to create a darkening of the gap between workpieces during the movement of the tool. The proposed algorithm in this paper is not a complex scheme consisting of several consecutive operations. The input data represent a restricted area of interest containing the front gap relative to the welding point. The image of the area of interest is segmented by using a Canny operator. A Hough boundary detector is applied to the binarized image, creating candidate lines, after which false gaps are filtered out by the Kalman operator. However, given the specifics of the Canny operator [25], the proposed algorithm will not allow for adaptive use. For various surfaces of welded workpieces, the Canny operator requires correction of configuration parameters, which makes it difficult to use the proposed algorithm in the developed automated tool correction system (LRC-W).

In [26], a method for determining control points of correction is considered, but its applicability is limited to a small neighborhood of the current position of the LRC-W tool, since it implies a local approximation of the trajectory by a mathematical function.

Detecting the gap using contrasting brightness segmentation in conditions of good illumination of the workpiece surface is the most promising way to highlight the desired object in the image. Given the basic requirement of laser welding to find focused radiation at the central point of the gap [27], detecting the gap as a set of the darkest pixels, with sufficient illumination, can significantly reduce the requirements for recognizable edges, since they can act as the boundaries of the dark area of the gap, which in most cases coincide with the edges of the gap. This feature can be achieved by using a bilateral LED illumination that creates a shadow in the gap. In other words, segmentation of dark pixels does not require preparation of joint edges for recognition. In addition, the requirements for the surface texture of workpieces can be reduced. Recognizing and determining the center point of a dark line that represents a gap allows us to ignore mirror reflections and surface scratches, except for the presence of serious defects that are comparable in depth and length to the recognized gap. However, it remains a question of unambiguously determining the desired object in an image containing areas that fall within the range of the threshold selection in brightness but are not gaps.

In addition, in the analyzed works, the question of determining the correct position on the line relative to the current position of the LRC-W instrument remains open. The issue of transferring the corrected position of the tool from the two-dimensional image space to the working space of the LRC-W, considering its orientation deviated from the neutral position, remains to be formalized.

Correction of the trajectory points for the laser welding preparation process requires not only the implementation of the tool position correction but also the correction of the focal length $f$ of the laser radiation. Automation of laser focal length correction can be combined with automated focusing of the CCD camera, determined by the $f_c - f = 30$ mm difference (Figure 3).

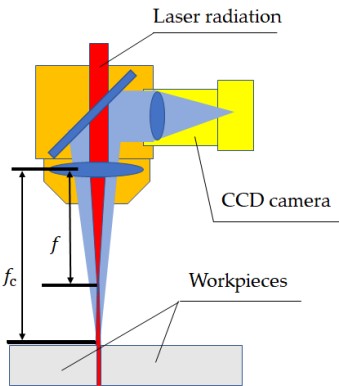

**Figure 3.** Scheme for determining the focal length of laser radiation and CCD camera.

Based on the analysis of the current state of the methods of automated correction of the LRC-W tool, the sequence of operations of the correction system is represented by the scheme shown in Figure 4.

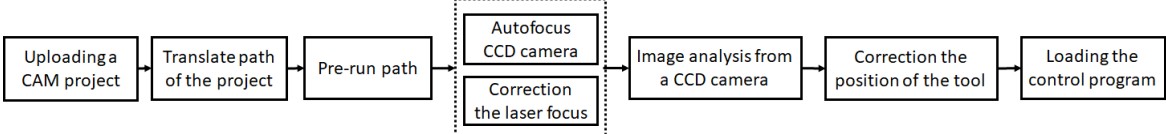

**Figure 4.** Scheme of the sequence of operations of the system for correcting the points of the planned trajectory in the preprocessing of laser welding using computer vision systems.

Based on [28], the maximum value of the joint misalignment to obtain a high-quality result is < 1 mm. Additional design of the tool orientation at the points of the planned trajectory for the correction process in conjunction with bilateral illumination, in conditions of the joint misalignment not exceeding 1 mm, allow gap recognition.

The structural scheme of the equipment used in the LRC-W is shown in Figure 5. The laser equipment used makes it possible to achieve a focused beam diameter of 0.187 mm with a Rayleigh length of 9.617 mm and to provide welding of plates up to 10 mm thick with a gap width of up to 0.6 mm [29].

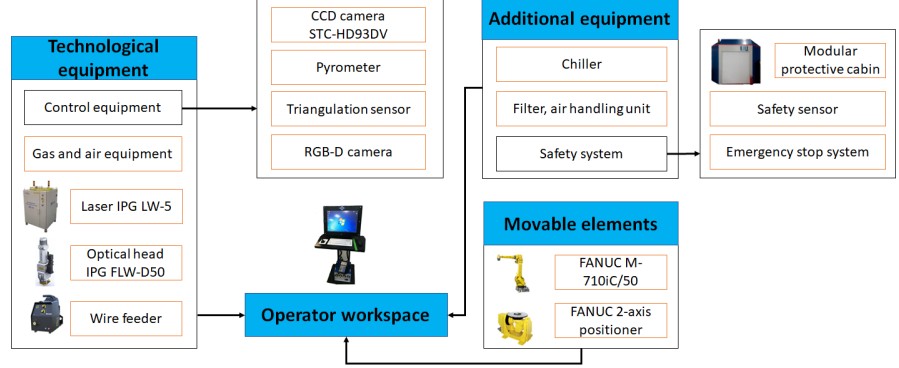

**Figure 5.** The structural scheme of a laser robotic welding complex (© 2020 IEEE [30]).

Preliminary research has shown that when the size of the area captured by the camera is $12.8 \times 7.2$ mm, it is possible to fully observe the darkened gap line with a width of 0.05 mm. The maximum deviation of the tool position from the gap line is estimated at 6 mm.

As a CAM/CAPP system, the author's development [31] is used, which provides technological preparation of the laser robotic complex (Figure 6).

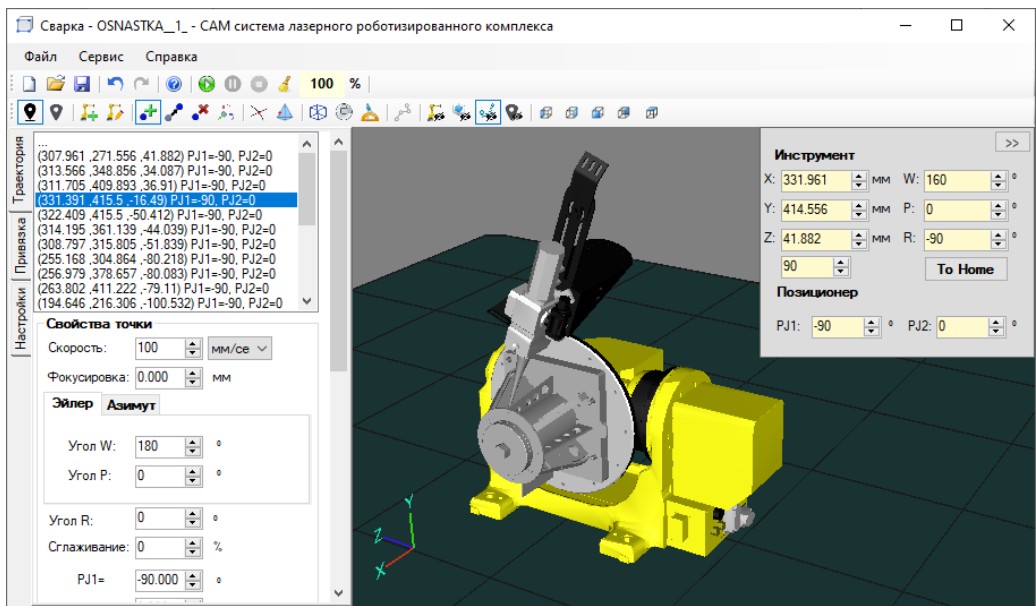

**Figure 6.** The authors' system for the technological preparation of a laser robotic complex.

The developed technological project in CAM is transmitted to the correction system, which is a module of the industrial LRC-W control system that interacts with technical equipment. As part of the correction system in the preprocessing of laser welding, a preliminary detour of the planned trajectory is performed and its correction is performed. After this, the control system sends the corrected control program to the robot controller (Figure 7).

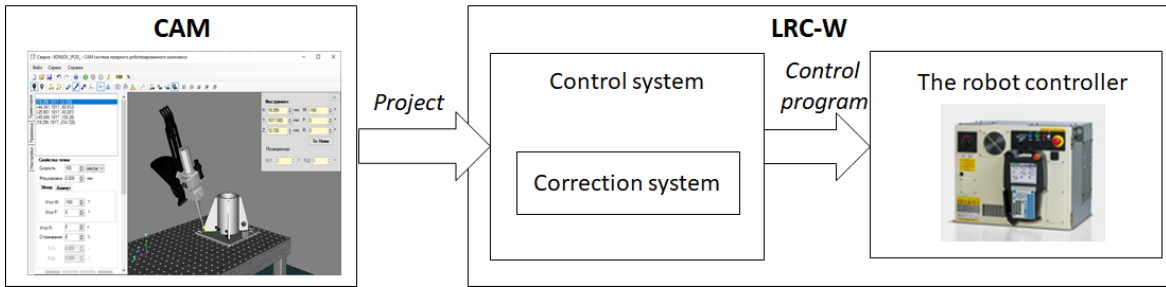

**Figure 7.** Scheme of the process of transferring the project from CAM to the LRC-W industrial complex using the trajectory correction system.

As a result, the aim of the research was to develop mechanisms for automating the correction of the focal length and position of the industrial LRC-W tool in the pre-process of laser welding, by processing images obtained from a CCD camera, allowing us to create and implement an automated system for correcting the points of the planned trajectory on an industrial LRC-W operating in small-scale and piece production.

To achieve this goal, the tasks are formulated, the solution of which forms the contribution of this paper:

1. Develop a method for jointly correcting the focus of laser radiation at the point of the planned trajectory and automating the focus of the CCD camera by moving the LRC-W tool along its own orientation axis;

2. Determine the dependence of the boundaries of the range of threshold pixel selection gap in the image, allowing for an unambiguous segmentation of the welded edges in the image obtained with the CCD camera tool LRK-W;

3. Carry out the formalization of the recognition of the welded edges of the workpiece on the result of segmentation, considering the presence of image regions in the rising range threshold selection for brightness, but not the gap;

4. Based on the result of edge recognition, determine the correct position of the point of the planned trajectory of the LRC-W tool;

5. Formalize the transfer of the corrected point value of the planned trajectory from the flat image space to the working space of the LRC-W, considering the current orientation of the tool.

## 3. Joint Correction of the Focal Length of Laser Radiation and Focusing of the CCD Camera

It is possible to correct the focal length of laser radiation by focusing the image of a CCD camera. Then, based on the scheme shown in Figure 6 and the assessment of the degree of image clarity when moving the tool along the orientation axis, it is possible to determine the focused image and therefore the focused position of the laser.

In [32], we consider various methods for evaluating the image clarity obtained from a CCD camera. As the most promising method applicable to the correction system, it is worth highlighting methods based on calculating the image gradient. The main advantages of the gradient method are high calculation speed and efficient application in a small image capture area with contrasting transitions in the focused position.

The definition of the value based on which the image clarity is evaluated in this case will be represented by the following sequence of actions:

- Gaussian blur $P_g$ of the original image $P$;
- Calculating the grad $P_g$ gradient;
- Calculating the sum of the pixel brightness of the blurred image gradient $P$ in the Hue, Saturation, Value (HSV) space, $S(P) = \sum I(\text{grad}P_g)$, where $I(\text{grad}P_g)$ is the functional brightness matrix of a flat image.

In the process of obtaining images $P_i$ from a CCD camera, the focused image $P^*$ is determined by the extreme value $P^* = \{P^* \in P_i \text{ and } S(P^*) \rightarrow max\}$.

The focused image is determined during the movement of the LRC-W tool along the localization area starting from the position $\overline{p}_{max}$ and ending at $\overline{p}_{min}$, where the position $\overline{p}_{max} = -h{\cdot}\overline{n} + \overline{p}_{cam}$, $\overline{p}_{min} = h{\cdot}\overline{n} + \overline{p}_{cam}$, $\overline{p}_{cam}$ is the position of the LRC-W tool positioning relative to the camera focus point, $h$ is the value that characterizes the length of the localization area of the focused image and $\overline{n}$ is the normalized vector describing the orientation of the LRC-W tool (Figure 8).

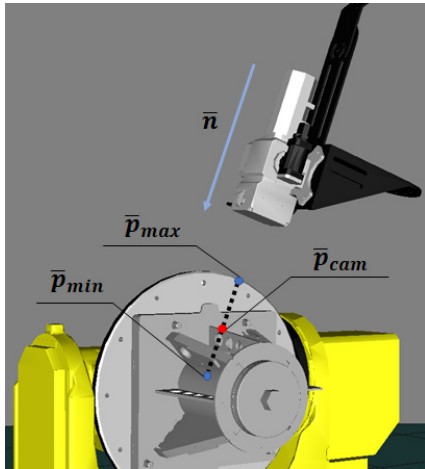

**Figure 8.** Scheme of tool movement in the process of automated detection of the focused position of the CCD camera.

When estimating the time of calculating the value of $S(P)$ in $t_0$, the speed with the minimum absolute error $\delta$ will be estimated as:

$$v = \frac{\delta}{t_0}.$$

Let the position of the $p^*_{cam}$ tool relative to the focus of the CCD camera be the focused position of the observed objects with focus $f_c$; then, the focused position of the $p^*_{tool}$ tool relative to the laser radiation with focus f is defined as:

$$\overline{p}^*_{tool} = (f_c - f) \cdot \overline{n} + \overline{p}^*_{cam}$$

Images from a CCD camera captured during autofocus are shown in Figure 9. The results of the image definition assessment are shown in Figure 10.

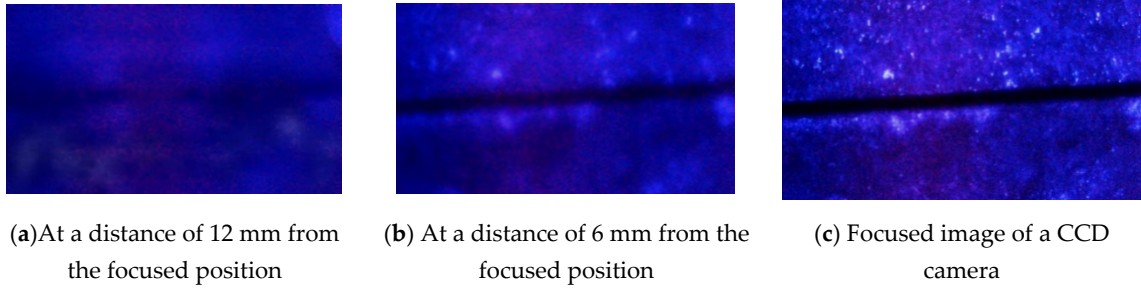

(**a**)At a distance of 12 mm from the focused position     (**b**) At a distance of 6 mm from the focused position     (**c**) Focused image of a CCD camera

**Figure 9.** Images captured from a CCD camera during autofocus.

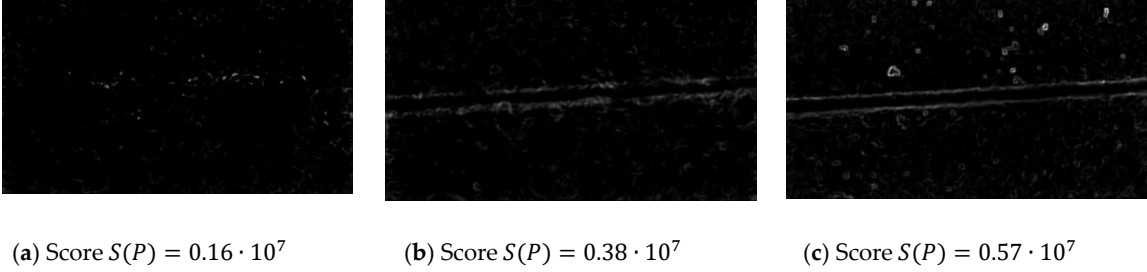

(**a**) Score $S(P) = 0.16 \cdot 10^7$     (**b**) Score $S(P) = 0.38 \cdot 10^7$     (**c**) Score $S(P) = 0.57 \cdot 10^7$

**Figure 10.** Result of image blur estimation.

Thus, a method was developed for jointly correcting the focus of laser radiation at the point of the planned trajectory and automating the focus of the CCD camera by moving the LRC-W tool along its own orientation axis.

## 4. Segmentation and Recognition of the Gap to Be Welded

Assessment of gaps between welded workpieces is performed based on the result of brightness segmentation of the image. To do this, one can take advantage of the fact that when there is sufficient light in the image, the gap is represented by a continuous dark stripe (Figure 11).

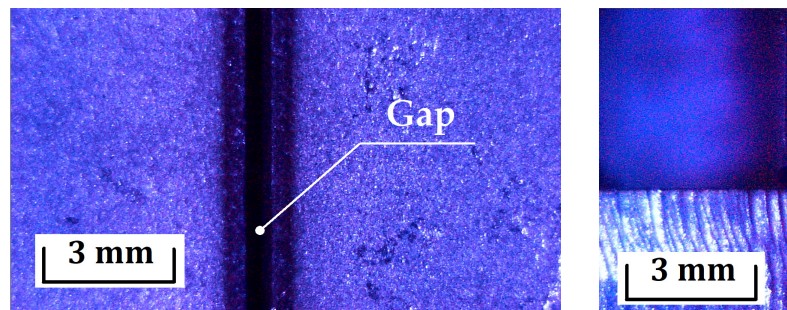

(**a**) Part gap line along the frame.

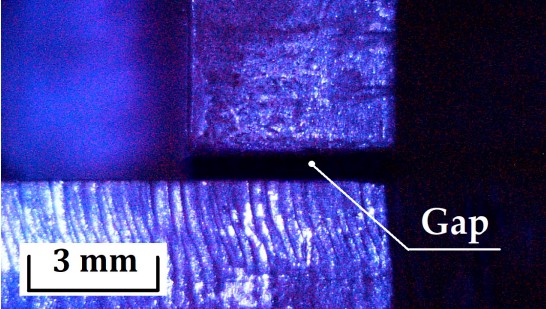

(**b**) Full gap line in the frame.

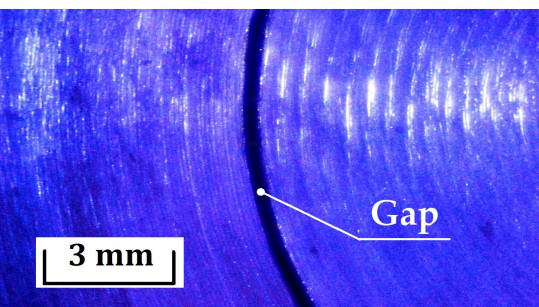

(**c**) Arc-shaped gap line in the frame.

**Figure 11.** Images of gap lines between welded workpieces with the LRC-W complex obtained from a CCD camera of HD resolution.

The images of the welded gap presented in Figure 11 were obtained in HD resolution from a STC-HD93DV CCD camera installed on the LRC-W tool according to the scheme shown in Figure 6. The system described in [20] was used as LED illumination. Figure 11a shows a photo containing part of the gap (the line continues beyond the photo), Figure 11b shows a photo containing the entire gap (including the edges of the line) of the workpieces being welded, and Figure 11c shows an image of an arc-shaped gap.

For line segmentation, the brightness range is determined and threshold selection is performed. However, welded metal surfaces that are illuminated with LED illumination to reduce the noise level in the image may have different reflection coefficients depending on the product and, as a result, a different range of pixel selection.

To determine the range, the distribution of image pixels in the HSV [33] space is calculated by the brightness value (Figure 12), where brightness is represented by a value in the range from 0 to 255 (0 is low brightness, and 255 is high brightness).

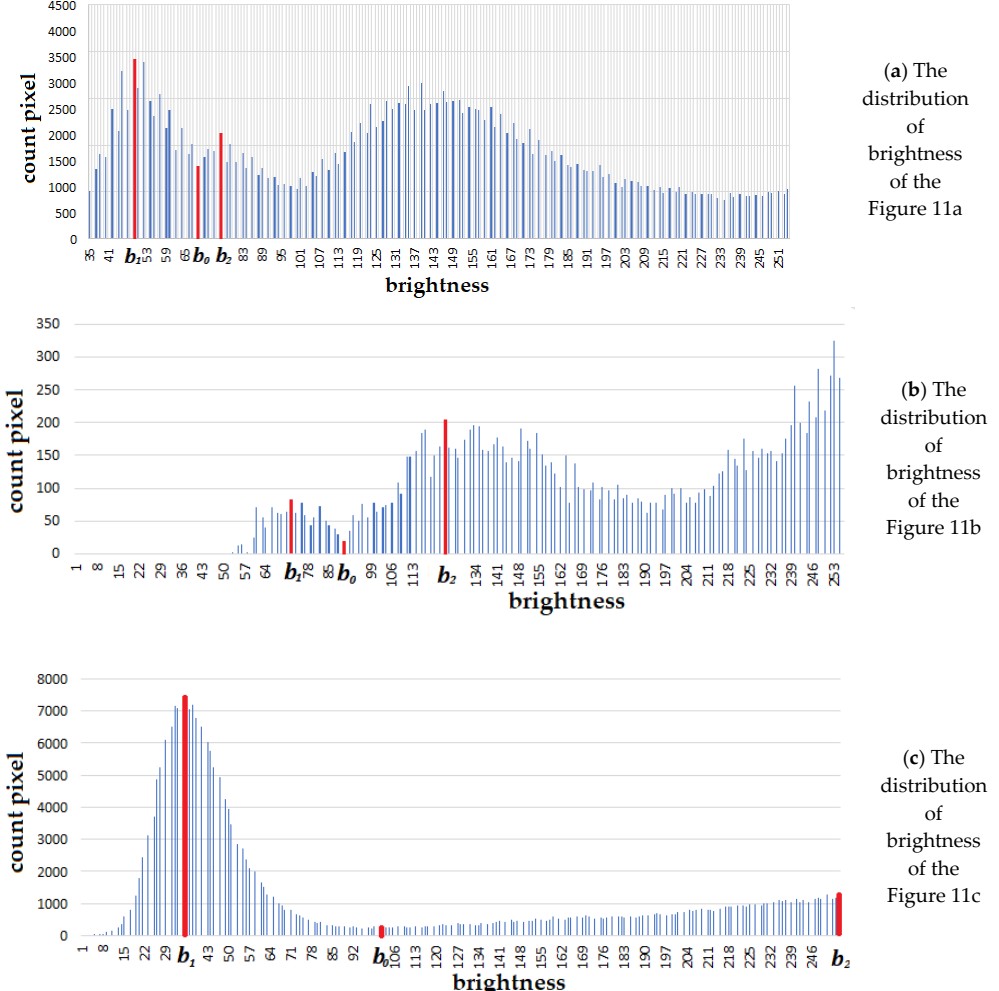

**Figure 12.** Histograms of the brightness distribution of pixels in the HSV space of the images in Figure 11 respectively.

The distribution graph shows that the images have several local maxima, $b_1$ and $b_2$, which characterize the dominant value of the brightness of the gap and the metal surface of the welded workpieces. The pixels of the gap are determined by belonging to the segment $[0, b_0]$, where $b_0$ is the point of the local minimum on the interval $(b_1, b_2)$. If there are a large number of local maxima, $b_1$ and $b_2$ are the first two that have minimum brightness values.

During the analysis of the histogram, it was found that the vast majority of pixels have a maximum brightness of 255 in the HSV space. Since the number of pixels with the maximum brightness in the research is not interesting, the value is cut out of the distribution.

Segmentation of the gap on the original image obtained from the CCD camera is represented by the following sequence of actions:

1. Grayscale translation of the original image in the HSV color space;
2. Lightening the image 1.5-fold;
3. Threshold selection (including binarization) of an image by brightness from 0 to $b_0$;
4. Applying the close morphological transformation (one of the combinations of erosion and dilation transformation) with a $3 \times 3$ core size;
5. Calculating a gradient with a $2 \times 2$ core size.

The result of the segmentation of the original images in Figure 11 is shown in Figure 13.

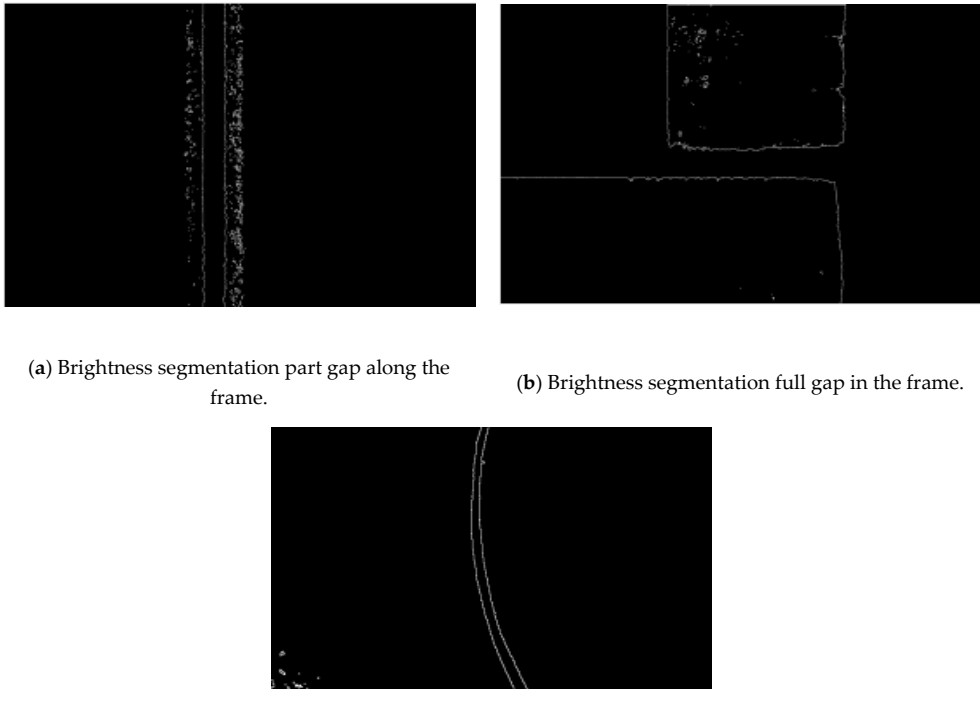

(**a**) Brightness segmentation part gap along the frame.

(**b**) Brightness segmentation full gap in the frame.

(**c**) Brightness segmentation arc-shaped gap in the frame.

**Figure 13.** The result of brightness segmentation of the gap of the welded workpieces.

Thus, the boundaries of the threshold selection in the process of segmentation of the welded edges of the blank gap were determined by the extreme values of the pixel distribution of the entire image in brightness.

The edges of the gap between the welded workpieces are characterized by two conditionally parallel lines. Their recognition, based on segmentation results, is performed using contour analysis tools.

A contour in the computer vision is an external boundary that describes the outline of a geometric object in the image [34]. The process of obtaining all the contours $C = \{C_i\}_{i=1}^{N}$ ($N$ is the number of contours) of an image can be performed using standard computer vision libraries (for example, OpenCV). The result in this case is an array of contours, each of which is represented by a set of segments $C_i = \left\{\left[a_{ij},\ b_{ij}\right]\right\}_{j=1}^{N_i}$, where $N_i$ is the number of segments i of the contour defined by two points, $a_{ij}$ and $b_{ij}$, on the plane.

To determine the pairs of $X$ and $Y \in C$ contours characterizing the edge line of the gap of the welded blanks, we used the following conditions:

$$\exists! X \text{ and } Y : \| Y \| - k_0 \cdot \| Y \| < \| X \| < \| Y \| + k_0 \cdot \| Y \|, \tag{1}$$

$$\forall C_i,\ \exists! X \text{ and } Y : \| C_i \| < \| X \| - k_1 \cdot \| X \| i \ \| C_i \| < \| Y \| - k_1 \cdot \| Y \|, \tag{2}$$

$$\mu(X,\ Y) = \sum_k \frac{\left| m_k^X - m_k^Y \right|}{\left| m_k^Y \right|} < \mu_0, \tag{3}$$

where $X = \max(\| C_i \|)$, $\| \|$ is the length of the contours, $m_k^X = sign\left(h_k^X\right) \cdot \log\left(h_k^X\right)$, $h_k^X$ is the hu moment, and $m_k^Y$ is calculated similarly to $m_k^X$.

Equation (1) defines the proximity of contours $X$ and $Y$ of length with the proximity coefficient $k_1$. Equation (2) selects the contours $X$ and $Y$ in length among all $C$ with the selection coefficient $k_2$. The moment of Equation (3) determines the statistical similarity of the contours, with the similarity parameter $\mu_0$.

If $X \cup Y = \varnothing$ or $|X \cup Y| > 2$, where $|\ |$ is the power of the set, the segmentation result contains large interference or false geometric outlines similar to the edges of the joint.

The solution to the problem of recognizing the edges to be welded requires expansion in the case when the image contains the edges of the gap that pass into the external borders of the workpieces. The need to expand the recognition task is related to the fact that the transition point can be the beginning or end point of laser welding (Figure 14).

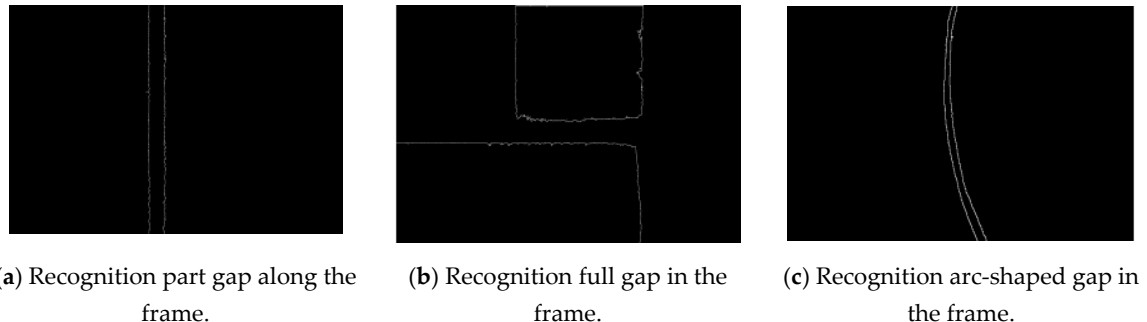

(**a**) Recognition part gap along the frame.

(**b**) Recognition full gap in the frame.

(**c**) Recognition arc-shaped gap in the frame.

**Figure 14.** Result of recognition of the gap based on segmentation results.

Defining the sections $X_0 \subseteq X$ and $Y_0 \subseteq Y$ that represent the edges of the joint uses the attribute that characterizes them. Its essence lies in the fact that, locally, $X_0$ and $Y_0$ are conditionally parallel contours (a slight deviation of the segments in the direction relative to each other is permitted, which allows them to be considered parallel).

Since the contours $X$ and $Y$ can consist of a large number of segments, they are reduced by applying the Douglas-Pecker algorithm to reduce the number of calculations [35].

The determination of segments belonging to the edges of the joint of the welded workpieces is carried out by the method of sorting and pair comparison of elements $X$ and $Y$. In this case, the segments $[a_x, b_x] \in X_0$ and $\left[a_y, b_y\right] \in Y_0$ form the edges of the joint if the following conditions are met:

$$\angle\left([a_x, b_x], \left[a_y, b_y\right]\right) < \varphi, \tag{4}$$

$$\left(\left[a_x, b_x^+\right] \cup \left[b_x, b_x^+\right] \cup \left[a_x^+, b_x^+\right] \cup [a_x, b_x^-] \cup [b_x, b_x^-] \cup [a_x^-, b_x^-]\right) \cap \left[a_y, b_y\right] \neq \varnothing, \tag{5}$$

where $\angle$ is the angle between the segments, $b_x^+ = w \cdot \bar{n} + b_x$, $b_x^- = w \cdot (-\bar{n}) + b_x$ ($a_x^+$ and $a_x^-$ similarly), $\bar{n}$ is defined as $\bar{n} \perp [a_x, b_x]$, and $w$ is the coefficient that characterizes the width of the gap.

If the set in Equation (5) is not empty, then verification of ownership is performed via $a_x \in P$ or $a_y \in P$, where $P$ is a closed area bounded by segments $[b_x^-, b_x^+]$, $[b_x^+, a_x^+]$, $[a_x^+, a_x^-]$, $[a_x^-, b_x^+]$ (Figure 15).

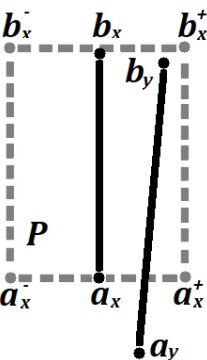

**Figure 15.** Scheme for determining the segments that belong to the edges of the joint and the parts to be welded (© 2020 IEEE [5]).

It is worth noting that the values of $a_x$, $b_x$, $a_y$, and $b_y$ are two-dimensional vectors of the flat image space.

The resulting sets of segments $X_0$ and $Y_0$ represent the edges of the workpiece gap to be welded. An example of the recognition result is shown in Figure 16.

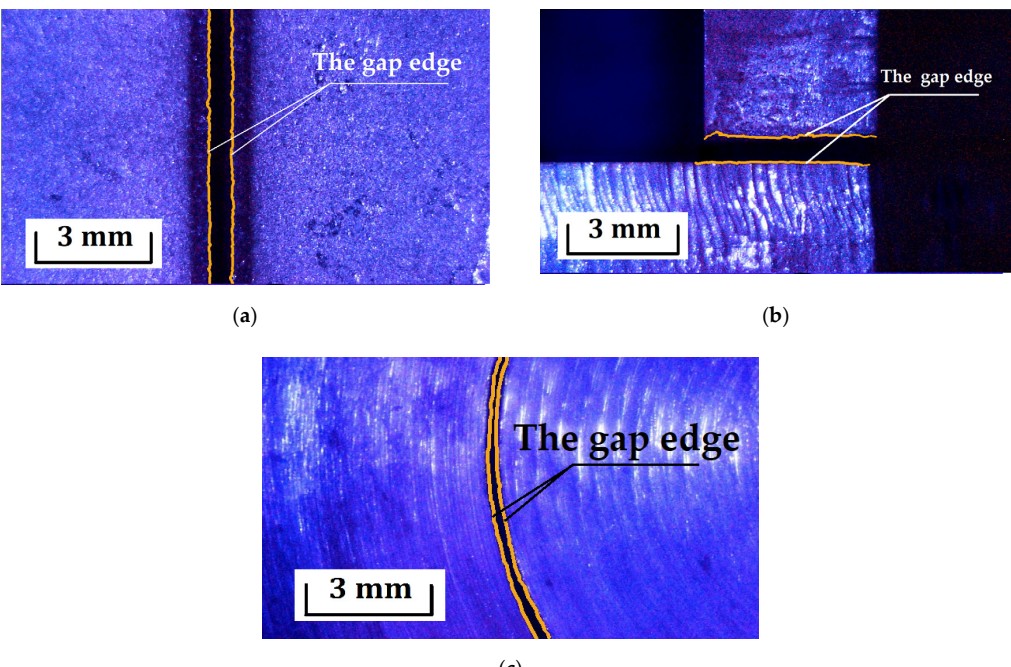

**Figure 16.** The result of recognition of the gap of welded workpieces in the image obtained from the CCD camera LRC-W with used value parameter recognition: (**a**) w = 65 $\varphi = 5$, $\mu_0 = 0.4$, $k_0 = 0.1$, $k_1 = 0.7$; (**b**) w = 55 $\varphi = 8$, $\mu_0 = 3.5$, $k_0 = 0.5$, $k_1 = 0.5$; (**c**) w = 30 $\varphi = 11$, $\mu_0 = 0.4$, $k_0 = 0.1$, $k_1 = 0.7$.

In this way, the rules and parameters for recognizing welded edges based on the segmentation result were formalized as a pair of continuous, conditionally parallel lines.

## 5. Determining the Correct Position of the LRC-W Tool and Translation of the Coordinates from the Image Space to the LRC-W Workspace

Correction of the position of the LRC-W tool at the point of the movement path should be carried out when the current position of the $p_{tool}$ is not located inside the welded gap between workpieces.

The main candidates for the correct position of $p^*$ points $p_{tool}$ relative to the gap are:

1. The starting/ending point of welding $a_0$, defined by the expression $a_0 = \frac{a_1 + a_2}{2}$, where points $a_1$ and $a_2$ are the extreme positions of the contours $X_0$ and $Y_0$ (not counting the intersection with the image borders);

2. The closest point $b_0$ of the gap defined by the expression $b_0 = \frac{b_1 + b_2}{2}$, where

$$b_1 = \{b_1 \in [a, b] \text{ and } [a, b] \in (X_0 \cup Y_0) : |b_1 - p_{tool}| \to min\}, \tag{6}$$

$$b_2 = ((X_0 \cup Y_0)/b_1) \cap L(b_1, d), \tag{7}$$

$$(b_1, n) \equiv (\frac{x - b_1}{n_x} - \frac{y - b_1}{n_y} = 0). \tag{8}$$

The line $L(b_1, n)$ is a straight line on a plane passing through the point $b_1$ in the direction of $n$ (here, $n \perp [a, b]$).

The correct position of the tool is defined as $p^*_{tool} = a_0$ if there are welding start/end points on the image, and $p^*_{tool} = b_0$ in all other cases. The result of applying the correction point calculation method is shown in Figure 17.

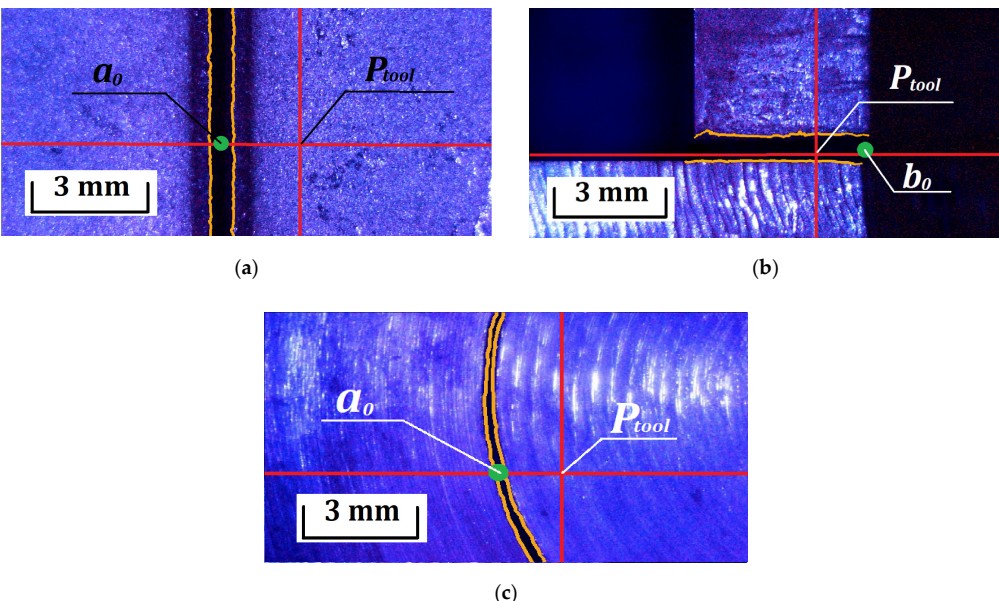

(a)

(b)

(c)

**Figure 17.** The result of determining the correction point in the case: (**a**) part gap along the frame; (**b**) full gap in the frame; (**c**) arc-shaped gap in the frame.

Relative to the correct start and end points of welding, sections of acceleration and deceleration of the tool are designed to ensure that the target speed is reached at the edges of the processing zone in order to reduce edge defects [36].

In this way, candidates for the correct position of the current position of the LRC-W tool relative to the recognized edges of the blank gap were identified.

The positioning of the LRC-W tool in the working space is described by the position of the laser focus $p_{tool}$ and the orientation of the laser beam, represented by the Euler angle system $\varepsilon = \{\varepsilon_x, \varepsilon_y, \varepsilon_z\}$ [37].

Image pixels are positioned relative to the upper-left edge with the axes pointing along the edges (Figure 18).

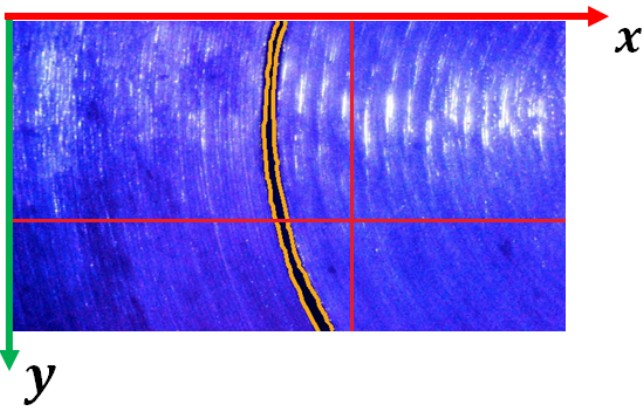

**Figure 18.** Image coordinate system.

The position of an arbitrary point $p'$ of a flat image is translated into the space of an industrial complex by the transformation:

$$p = s \cdot p' \cdot T, \tag{9}$$

where *s* is the scale factor and *T* is a transformation matrix defined as:

$$T = \begin{pmatrix} \vec{i}'_x & \vec{i}'_y \\ \vec{j}'_x & \vec{j}'_y \\ \bar{k}'_x & \bar{k}'_y \end{pmatrix}.$$ (10)

The $\vec{i}'$, $\vec{j}'$, and $\bar{k}'$ axes (Figure 19) are defined as:

$$\vec{i}' = R_z(\beta_z - \Delta\varepsilon_z) \cdot R_y(\beta_y - \Delta\varepsilon_y) \cdot R_x(\beta_x - \Delta\varepsilon_x) \cdot \vec{i}$$ (11)

$$\vec{j}' = R_z(\beta_z - \Delta\varepsilon_z) \cdot R_y(\beta_y - \Delta\varepsilon_y) \cdot R_x(\beta_x - \Delta\varepsilon_x) \cdot \vec{j}$$ (12)

$$\bar{k}' = R_z(\beta_z - \Delta\varepsilon_z) \cdot R_y(\beta_y - \Delta\varepsilon_y) \cdot R_x(\beta_x - \Delta\varepsilon_x) \cdot \bar{k}$$ (13)

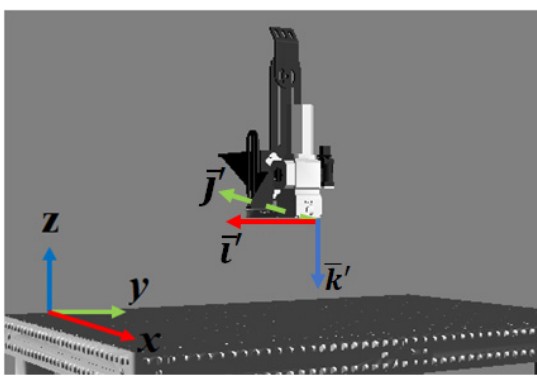

**Figure 19.** Location of the image base $\{\vec{i}', \vec{j}', \bar{k}'\}$ relative to the base of the LRC-W workspace.

Here, $\bar{i}$, $\bar{j}$, and $\bar{k}$ are the basis vector of the base of the LRC-W workspace; $R_x(\varphi)$, $R_y(\varphi)$, and $R_z(\varphi)$ of the rotation matrix by the angle $\varphi$ around $\bar{i}$, $\bar{j}$, and $\bar{k}$, respectively; $\beta = (\beta_x, \beta_y, \beta_z)$ is the angle vector that characterizes the location of the CCD camera's field of view in the LRC-W space with a neutral orientation $\varepsilon_0$, and $\Delta\varepsilon = \varepsilon_0 - \varepsilon$ determines the deviation of the tool with the current orientation $\varepsilon$ from the neutral $\varepsilon_0$.

In this way, the calculation model for transferring the correct position from the flat image space to the LRC-W workspace was formalized, considering the orientation of the tool and the camera position.

## 6. Discussion

The results of the study show that the developed mechanisms allow one to achieve automated correction of the points of the planned trajectory of the industrial LRC-W tool in the preprocessing, calculating the correct position relative to the welded edges of the gap between workpieces recognized in the image obtained from the CCD camera installed on the LRC-W tool.

To ensure that the gap to be welded can be recognized in the image obtained from the CCD camera, the research area is illuminated with LED illumination, which reduces the noise level. The bilateral placement of the LED backlight allows us to additionally darken the gap and create an area of dark pixels.

Deviations of the position of the LRC-W tool at the points of the planned trajectory based on the result of transfer from the CAM model space create the problem of a blurred image that does not allow recognition of the objects of interest. The problem is fixed by automatically focusing the CCD camera. The automatic focusing mechanism uses the tool's movement along its own axis orientation. In the process of focusing, the image clarity is evaluated using the gradient method, which is used to determine the most focused image. Using the known difference between the focus of the camera and

the focus of the laser radiation on the basis of the result of focusing the CCD camera, the current focal length of the laser radiation at the point of the planned trajectory is determined and corrected.

Recognition of the edges of the gap is performed by pre-contrast segmentation of the desired object by brightness. The boundaries of the threshold selection segment of pixels that characterize the desired area in the brightness range are the extreme values of the pixel distribution of the entire image in brightness. On the one hand, this method allows one to determine the desired gap and make the necessary corrections relative to it; on the other hand, it imposes additional requirements on the illumination of the study area so that the line represents the area of the darkest pixels of the image. Our research has shown that a qualitatively highlighted area of the image provides segmentation of the desired line.

As the intermediate results of the study show, the segmentation operation does not allow one to uniquely determine the desired edges of the gap, since the image may contain various elements located in the threshold selection area. This feature has been confirmed by other researchers. In this regard, the segmentation results are used to recognize the edges of the gap to be welded. The features that characterize the desired edges are a set of rules that correspond to the selection on the image of those segments that are conditionally parallel and located relative to each other within the specified width of the gap.

As candidates for the correct position of the planned trajectory relative to the current position of the tool, the closest and two edge points of the gap are used. Thanks to technological solutions in the project, in most cases, it is possible to have a single gap in the image, relative to which the correct position is determined. However, the possibility of having multiple lines is not excluded. The developed methods allowed us to recognize several gaps in a single image, but the methods used to determine candidates for the correct position of the tool did not provide the possibility of technologically correct position determination. In this regard, the issue of developing rules for determining the correct position in cases where there are several gaps with different relative positions on the image in this work remains open and represents an opportunity for further work by the authors.

Translation of the coordinates of the correct position of the flat image space into the three-dimensional working space of the industrial LRC-W is carried out according to the developed calculation model. The model considers the position of the LRC-W tool and the location of the CCD camera in space, allowing for unambiguous determination of the correct position of the robotic arm in the world space at a given focal length.

The results obtained were applied in the authors' automated laser welding control system [38], for correcting the points of the tool trajectory transmitted from the CAM system [15] to the working space of the LRC-W. The solution was designed as an automated subsystem that functions in the preprocessing of laser welding and creates a preliminary detour of the trajectory and correction of its points relative to the gap.

During the primary experimental research of the automated subsystem, successful results of the correction subsystem functioning were obtained. Some of the results are shown in the figures in this paper. At the moment, the authors plan to develop a methodology for full tests to determine the conditions for limiting the functioning of the developed methods, which is a further task in their research work.

**Author Contributions:** Project administration, L.A.; conceptualization and methodology, R.D. and P.V.; software, R.D.; writing, R.D.; review and editing, R.D. and L.A.; validation, R.D., L.A., and P.V.; experimental research, L.A. and R.D. All authors have read and agreed to the published version of the manuscript.

**Funding:** This research received no external funding.

**Acknowledgments:** The research was carried out at the Scientific and Educational Center for the Introduction of Laser Technologies, Vladimir State University named after Alexander and Nikolay Stoletovs.

**Conflicts of Interest:** The authors declare no conflict of interest.

**Patents:** The results of the work were applied to the automated process control system for laser robotic processing, which received a certificate of registration (Patent RF 2020661002, 2020).

## Abbreviations

The following abbreviations are used in this manuscript:

| | |
|---|---|
| CAD | computer-aided design |
| CAM | computer-aided manufacturing |
| CAPP | computer-aided process planning |
| CCD | charge-coupled device |
| HSV | hue, saturation, value |
| LED | light-emitting diode |
| LRC-W | laser robotic complex-welding |
| TCP | tool center point |

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
