# Peer review of "Development of Mechanisms for Automatic Correction of Industrial Complex Tools in the Preprocessing of Laser Welding for Small-Scale and Piece Production Using Computer Vision†"

_machines, doi:10.3390/machines8040086_

Round 1

Reviewer 1 Report

The paper is generally well written and presented and could be accepted in the current form.

Author Response

Authors thanks so much for your review report.

Reviewer 2 Report

Dear authors, 

the topic is very interesting and actual. The paper addresses a topic which is within the journal s scope and uses relevant literature. 

Athough, the paper has reached a very good level, before publication, there are several aspects that need to be addressed, namely: 

please clarify the identified gap which you are aiming at fulfilling through your article, it lacks a scientific contribution.  In this part, which gap is filled in the scientific field, research field. 

Since the article will be published as scientific article, please clarify the contribution of the paper to enriching the state of knowledge and also consider enhancing the final part by providing more conclusions/implications/further directions of research etc.

Author Response

Please see details in attachment.

Reviewer 3 Report

The paper addresses the industrially relevant issue of joint finding in laser welding.

Focus in the presentation is on a very basic image feature extraction method and common coordinate transformations between world and image coordinates. All results presented are based on only two photographs of which no information about how they have been captured is given.

The paper does not present any requirements on the laser welding process or the image acquisition system so the generality of the suggested method is not possible to asses making it very questionable to the laser welding society.

The language used uses a number of nonstandard nomenclature making it hard to read.

There is no proper discussion and conclusions.

The authors claims that they have obtained successful results with no support from the presentation. This is something that is not OK.

Author Response

Please see details in attachment.

Reviewer 4 Report

The authors propose an automated system based on computer vision, which corrects the point of trajectory in the working space of the industrial Laser Robotic Welding Complex (LRC-W). According to the authors, the paper is an extension of their previously published work. 

I have the following suggestions:

  1. The Abstract requires major revision and editing. The flow must be improved. One example: this following line is in the Abstract, which does not make sense to me:
    " As a correction method using the method of recognizing the edges of the separating line to be welded on the image obtained from a CCD camera installed on an industrial complex tool."
  2. The introduction needs to be significantly modified to establish the motivation as well as contribution more clearly.
  3. The paper does not discuss the background or related work clearly/elaborately. The authors mention 3 generic shortcomings of existing methods based on ref 16- 25. However, it is not clear which referred methods correspond to which shortcomings. A better comparison between existing works and the proposed method would be helpful for the readers to understand the advantage/significance of the proposed method. 
  4. First appearance of acronyms should be expanded. For example, several acronyms are used in the paper(CAM, CAPP), full-forms of which are not mentioned anywhere.
  5. In Figure 3, the difference between 'part separation' and 'full separation' is not clear to me.
  6. The discussion section is under-explained. The authors claim that "The results obtained were applied in the author's automated laser welding control system". Proper explanation of the whole testing procedure would help the readers to understand the performance. However, no details are provided in the paper.
  7. I went through the previous work of the author: "Dmitry Rodionov, Lyukhter Alexander, Prokoshev Valery, Methods of Automatic Correction of Technological Trajectory of Laser Welding Complex by Means of Computer Vision”, 2020 IEEE International Conference on Industrial Engineering, Applications and Manufacturing (ICIEAM), 18–22 May 2020."

    I would like to know from the authors: what are the new inclusions to this paper? I am unsure about the percentage of new content in this paper for another publication. Authors' justification would help me clarify my doubts.
  8.  The paper has major grammatical mistakes throughout. There are multiple sentences that are formed incorrectly. The organization of the paper requires significant improvement. Extensive editing of English language and style required.

Author Response

Please see details in attachment.

Reviewer 5 Report

Multiple-citations like [1-4], [5-7], [10-13] or [16-25] must be avoided, so that each statement can be assigned to a specific reference.

The statement with respect to [1-4] would probably fit on thousands of papers.

Abbreviations must be introduced in the text at first use, also CAM and CAPP.

A scientific state of research was not described.

In my understanding, the paper is more a technical report on an implementation technique.

Author Response

Please see details in attachment.

Round 2

Reviewer 3 Report

The manuscript has been improved regarding the language and clarity in the disposition. The discussion is now a little bit more convincing and the conclusions slightly better supported by the results. However, I still suggest some major revisions.

The paper lacks necessary information about the application addressed by the suggested mechanism, information that is needed to support the validity of the results and to assess the generality of the method.

  • What is the laser beam diameter in focus and what is the Rayleigh length? This information have a huge impact on the sensitivity to e.g. the joint gap width and the joint centre line and laser beam offset.
  • What are the tolerances to:
  • joint gap width,
  • joint misalignment,
  • laser beam focus position,
  • joint edge preparation?

All this information will influence the requirements on the joint finding system.

  • What is the plate thicknesses in the targeted application? This information will define the admissible gap bridging.
  • What are the tolerances on joint edges? This will for sure influence the line detection.
  • Is it only butt joints that are addressed?
  • What are the requirements on surface texture and conditions such as absence of scratches and specular reflections? These factors will have a huge impact on the robustness of the algorithm not at least on how the led illumination has to be directed onto the area of interest. For example machined surfaces of titanium alloy Ti-64 display very high reflectivity making it much trickier to get sufficient information about edges and surfaces.
  • What is the depth of focus obtained in the image acquisition? If the z-offset is changing this will for sure influence the image quality and further the feature extraction.

Author Response

Please see details in attachment.

Reviewer 4 Report

I would like to thank the authors for making necessary changes as per the suggestions provided to them. I believe, the paper is now well-organized and better presented than the previous version. The motivation is well-explained.

I wanted to know in my previous report: How does the work differ from the authors' previous work ("Dmitry Rodionov, Lyukhter Alexander, Prokoshev Valery, Methods of Automatic Correction of Technological Trajectory of Laser Welding Complex by Means of Computer Vision”, 2020 IEEE International  Conference on Industrial Engineering, Applications and Manufacturing (ICIEAM), 18–22 May 2020); or what are the new inclusions to this paper that has been submitted. 

The authors provide an explanation in the cover letter. Majority of the new inclusions specified by the authors are organizational, but not technical. The new inclusions are mostly in terms of better explaining the method, including extensive reviews of previous works, putting better images, modifications on introduction, elaborated discussions and so on. However, I am not convinced that the percentage of new work included in this manuscript from technical viewpoint is sufficient for another publication. 

From my understanding, the new technical inclusions to this paper are generalizing and parameterizing the edge detection method, and presenting a mathematical method for transferring the correct position of the tool from a flat image space to the working space of an industrial complex. 

Author Response

Please see details in attachment
